# Multiple Head Rotations Result in Persistent Gait Alterations in Piglets

**DOI:** 10.3390/biomedicines10112976

**Published:** 2022-11-19

**Authors:** Mackenzie Mull, Oluwagbemisola Aderibigbe, Marzieh Hajiaghamemar, R. Anna Oeur, Susan S Margulies

**Affiliations:** 1Wallace H. Coulter Department of Biomedical Engineering, Georgia Institute of Technology and Emory University, Atlanta, GA 30322, USA; 2Department of Biomedical Engineering and Chemical Engineering, University of Texas at San Antonio, San Antonio, TX 78249, USA

**Keywords:** concussion, gait, pediatric, swine, rapid non-impact head rotation

## Abstract

Multiple/repeated mild traumatic brain injury (mTBI) in young children can cause long-term gait impairments and affect the developmental course of motor control. Using our swine model for mTBI in young children, our aim was to (i) establish a reference range (RR) for each parameter to validate injury and track recovery, and (ii) evaluate changes in gait patterns following a single and multiple (5×) sagittal rapid non-impact head rotation (RNR). Gait patterns were studied in four groups of 4-week-old Yorkshire swine: healthy (*n* = 18), anesthesia-only sham (*n* = 8), single RNR injury (*n* = 12) and multiple RNR injury (*n* = 11). Results were evaluated pre-injury and at 1, 4, and 7 days post-injury. RR reliability was validated using additional healthy animals (*n* = 6). Repeated mTBI produced significant increases in gait time, cycle time, and stance time, as well as decreases in gait velocity and cadence, on Day One post-injury compared to pre-injury, and these remained significantly altered at Day Four and Day Seven post-injury. The gait metrics of the repeated TBI group also significantly fell outside the healthy RR on Day One, with some recovery by Day Four, while many remained altered at Day Seven. Only a bilateral decrease in hind stride length was observed at Day Four in our single RNR group compared to pre-injury. In sum, repeated and single sagittal TBI can significantly impair motor performance, and gait metrics can serve as reliable, objective, quantitative functional assessments in a juvenile porcine RNR TBI model.

## 1. Introduction

Traumatic brain injury (TBI) is a leading cause of morbidity and mortality among children in the United States [1]. Globally, the estimated annual incidence of pediatric TBI ranges between 47 and 280 per 100,000 children, with the United States estimating about 70 per 100,000 children [2,3]. Between 1997 and 2017, there were over 95,000 TBI-induced deaths in children, known to be largely caused by motor vehicle accidents, child abuse, and falls [1,4]. In the developing brain, the extent, location, and mechanism of injury can cause poor neurological outcomes and functional disabilities that impact the somatic, cognitive, and emotional aspects of a child’s life [5,6]. Many studies on children have focused on the cognitive or behavioral changes caused by TBI, resulting in limited focus on the functional outcomes of gait velocity and balance post-injury.

Although persistent motor impairments in children have been reported after moderate to severe TBI [7,8,9,10], children with mild TBI also exhibit slower gait velocity or dual-task gait impairments that persist even after concussion symptoms have resolved [11,12,13], perhaps due to their efforts to avoid falling and to maintain stability [14]. Furthermore, TBI in younger children affects the developmental course of motor control [15].

Younger children are also known to have increased risk of sustaining repeated/multiple mild brain injuries [16,17,18]. Between 2002 and 2006, 51% of brain injuries reported each year occurred during the period of cerebral development (ages 0–24 years) and the estimated incidence of repeated injuries for this population ranges from 5.6% to 36% [4,19]. Repeated brain injuries in younger children are usually attributed to child abuse or falls [4,20,21]. Slower recovery in balance deficits, increased difficulties in memory and concentration, as well as increases in learning disabilities have been observed in adult and junior athletes who experienced multiple brain injuries [22,23,24]. However, the effects of repeated brain injury on gait are understudied in the pediatric population.

Animal models are essential in better understanding and treating motor impairments after TBI. An important consideration when choosing an appropriate animal model is replicating the injury type, neuropathology, and mechanisms in human TBI and applying proper biomechanical loadings to the head that can cause brain tissue deformations in animal models similar to humans. Although studies have utilized small animal models (i.e., rodents) to evaluate the impact of injury severity on histopathological changes and motor impairments, many of these animal studies have been unable to efficiently mimic the biomechanics of TBI observed in children due to differences in skull thickness, brain anatomy, and physiology. Additionally, scaling inertial and impact kinematics from adults to children are inaccurately captured because adult skull and brain properties cannot be linearly extrapolated to represent the infant and child head [25,26,27,28]. In addition, when compared to the human brain, the rodent brain is smaller in size, smooth, and possesses a lower white-to-grey matter ratio [29,30]. These structural differences may be responsible for the substantially different responses to trauma between rodents and humans [31,32]. It may also contribute to the failure of clinical trials for neuroprotective drugs that were identified as being effective in rodent TBI models [31]. In contrast, piglets are popular large animal models that are similar in brain anatomy, physiology, and development to children [33,34,35]. Compared to children’s brains, piglet’s brains have similar patterns of post-natal neurogenesis, similar time course of myelination, and similar white matter volume [36,37,38,39,40]. These similarities make piglets an appropriate animal model for evaluating and studying TBI in the pediatric population. Many gait studies in piglets have utilized focal injury models like the controlled cortical impact (CCI) model, which mimics focal contusions [38]. However, there are no piglet studies that have focused on the effect of diffuse TBI on gait. In this present study, we used a rapid non-impact head rotation (RNR) model that mimics the inertial diffuse injuries resulting from high translational and rotational accelerations of the head with or without impact [41]. These RNR injuries are usually caused by falls or low speed motor vehicle accidents, which account for most TBIs noted in young children [27,42]. Particularly, we concentrated on the sagittal RNR head movement known to injure the brainstem, which plays a crucial role in balance, posture, and locomotion [43]. In addition, most studies have utilized rodent animals to study the effects of single and repetitive brain injury on gait [25,44,45,46]. At this time, there are no studies that have evaluated the effects of repeated brain injury on gait in a large animal model.

Therefore, in this study, we used a piglet RNR model of TBI to study potential gait deficits due to single and repeated brain injuries in pediatric populations. We hypothesized that (1) gait time, velocity, cycle time, cadence, number of stances, stance time, and stride length are reproducible motor performance metrics in young pigs; (2) mild levels of rapid head rotations acutely affect gait; and (3) gait deficits are exacerbated with multiple head rotations. We studied four piglet cohorts: (i) healthy, (ii) anesthetized, uninjured sham, (iii) single RNR, and (iv) multiple RNR. Healthy animals were utilized to develop and validate performance reference ranges used as baselines for identifying important gait changes after TBI. These healthy and injured piglet data will provide a platform that can be used in the future to evaluate the influence of therapeutics and interventions on motor function following single and repeated TBI.

## 2. Materials and Methods

### 2.1. Animals

Forty-two 18–19-day old female and two 18–19-day old uncastrated male Yorkshire swine were received from Palmetto Research Swine (Reevesville, SC, USA), two 18–19-day old female Yorkshire swine were received from Oak Hill Genetics (Franklin County, IL, USA), and three 18–19-day old female Yorkshire swine were received from Premier BioSource (Ramona, CA, USA). Animals were given physical exams by the Emory University Division of Animal Resources (DAR) upon arrival to ensure no abnormalities were present, such as hoof deformations. Animals were received in cohorts of 2–3 littermates and housed together for the duration of the study. Housing consisted of a 12-h light-dark cycle with ad libitum access to pellets and water.

### 2.2. Accreditation

The protocol used in this research was approved by the Emory University Institutional Animal Care and Use Committee (IACUC). All lab space and animal records were inspected by the United States Department of Agriculture (USDA), the Association for Assessment and Accreditation of Laboratory Animal Care (AAALAC), and the Emory University IACUC.

### 2.3. Acclimation

Prior to data collection, animals were acclimated to research staff and equipment for a minimum of three days. A Tekscan Strideway™ pressure system (Tekscan Inc., MA, USA) was used to assess gait. The total area of the mat was 10.7 feet by 3.0 feet; the total area of active sensitivity was 6.4 feet by 2.1 feet. On the first day of acclimation, animals were exposed to the Tekscan Strideway™ mat as a cohort. Animals were placed on one end of the mat and encouraged to ambulate to the opposite end through the presentation of an auditory stimulus (e.g., a clicker). After successfully reaching the end of the gait mat, animals were rewarded with food enrichment (e.g., yogurt on a tongue depressor). This was repeated several times to allow for learning of the behavior as well as modelling the behavior to observing littermates. On the second and third day of acclimation, animals were exposed individually to the Tekscan Strideway™ mat using the same techniques described above.

### 2.4. Design of Animal Experiments Based on On-Field Head Impact Measurements in Soccer

The piglet TBI experiments in this study were designed based on the head impact kinematics that occur in high school soccer games. Measurements of video-confirmed frontal head-ball impact header kinematics during two seasons of high school competitions (rotated dominantly in sagittal plane) [47] were selected for the purpose of this study. Fortunately, the sagittal plane cerebrum kinematics have an anatomic fidelity between the quadruped and the biped. Also, our previous porcine studies showed that this direction can cause more severe axonal pathology compared to other rotational directions given the same peak head angular velocity [48]. The ball-head impact kinematic data from the high school athlete subjects (*n* = 267) were scaled for the pig TBI experiments in this study, as previously described in detail [49].

To summarize, the ratio of peak angular acceleration (α) to peak angular velocity (ω) (α/ω ratio) relates the frequency/duration of head impact rotational motion and represents the characteristics of head impact kinematics, and this ratio directly influences the intracranial axonal deformations [48]. Therefore, the α/ω ratio was calculated for each head impact (*n* = 267) from this high school data, and the 50th percentile of this ratio was calculated to represent the mean head impact kinematic characteristic in human soccer. The spectrum of diffuse brain injuries (including concussion, diffuse axonal injury, and coma [50]) result from the deformation of white matter tissues in the brain [51]. Therefore, the maximum axonal strain (MAS) value for each head impact (*n* = 267) was estimated using previously published MAS surface contours that relate MAS and head kinematics (peak angular velocity and peak angular acceleration) through brain finite element modeling [52]. Next, the 50th percentile and 90th percentile of MAS values in this human data were calculated and the intersection of these MAS curves with the 50th percentile α/ω were selected to represent the ‘median’ and the ‘high’ head kinematic loadings in humans. Then, a tissue deformation-based optimal scaling method [52,53] was used to identify scaled sagittal peak angular acceleration (α) and peak angular velocity (ω) values that produce the same MAS values in pigs as the ‘median’ and the ‘high’ head kinematic loadings in human soccer [53]. Following these steps, the peak angular velocity and acceleration that needed to be applied to the pig heads to replicate the ‘median’ and the ‘high’ head impacts experienced in human soccer games were found to be ω = 60 rad/s and α = 20–30 krad/s^2^ for the ‘median’ or 50th percentile, and ω = 100 rad/s and α = 50–60 krad/s^2^ for the ‘high’ or 90th percentile head kinematic loadings.

From the same on-field soccer head impact dataset [54], we evaluated the rate and interval between multiple headers on the field and found that at the 95th percentile, there were 6 and 4 impacts per hour for boys and girls, respectively, with an interval of 8 min. The most typical pattern for repeated impact was a single ‘high’, and 4 to 5 ‘median’ level loads. Therefore, our repeated head rotation group (multiple) received one ‘high’ followed by four ‘median’ level sagittal head rotations, spaced 8 min apart. The single head rotation group received one ‘high’ level rotation.

### 2.5. Head Rotation Methodology

A well-established, rapid non-impact rotational (RNR) injury model was used in this study to produce mild TBI in piglets similar to that of a sports-related concussion in adolescent humans.

Subjects were distributed into a naïve group and an experimental group. The naïve group had healthy animals with no anesthesia experience (*n* = 16, female; *n* = 2, male). The experimental group consisted of animals allocated to multiple RNR (*n* = 11, female), single RNR (*n* = 9, female), anesthesia-only shams (*n* = 8, female). The single RNR group experienced one ‘high’ level rotation followed by 32 min of anesthesia, and the repeated RNR group experienced one ‘high’ level rotation followed by four ‘median’ level rotations with 8 min intervals between rotations, totaling to 32 min of anesthesia. Anesthesia-only shams experienced no rotations and received 32 min of anesthesia. Within the naive group, a few animals were used to create a healthy reference range (*n* = 12, female), and the remaining were used to validate the created reference range (*n* = 4, female; *n* = 2, male). Healthy animals without an anesthetic experience were used to control for the effects of anesthesia while also establishing a healthy reference range. Healthy animals were studied for three non-consecutive days, and experimental group animals were studied for one day pre-injury and at 1, 4, and 7 days post-injury.

Prior to injury, animals were sedated with Ketamine (4 mg/kg), Xylazine (2 mg/kg), and Midazolam (0.2 mg/kg) via intramuscular (IM) injection. Animals were subsequently anesthetized with 5% isoflurane and 1.5–2.0 L/min of oxygen via gas mask. Once a surgical plane of anesthesia was achieved, characterized by a lack of toe pinch reflexes, the animal was intubated, placed on the ventilator (10–15 mL/kg), and secured to the pneumatic actuator by a padded snout clamp. Maintenance anesthesia was administered for the duration of the procedure at 3% isoflurane. Prior to the first rotational injury, Buprenorphine (0.1 mg/kg) was administered via IM injection and toe pinch reflex was checked again to confirm the depth of anesthesia. Isoflurane was then withdrawn, and the animal was removed from ventilator for less than 2 min. The head of the animal was rotated 60–70° in the sagittal plane at a target level of 100 rad/s (high level) over 15 milliseconds by inertial loading of the pneumatic device, with the center of rotation occurring in the cervical spine. Immediately post-injury, the animal was placed back on the ventilator with maintenance anesthesia provided. For single injury animals, ventilation and anesthesia were provided for 32 min post-injury. For multiple injury animals, ventilation and anesthesia were provided for 8 min and withdrawn again before the next rotation at a target level of 60 rad/s (low level) over 20 milliseconds. The low-level rotation was repeated four times, all occurring approximately 8 min apart with anesthesia being withdrawn prior to rotation. Actual sagittal angular velocity and acceleration are provided in Table 1.

After rotations and anesthesia were completed, all animals were checked for tongue lacerations, then removed from the ventilator and isoflurane. Once the animal was respiring independently and maintaining oxygen levels > 95%, the animal was extubated and transported to housing for recovery. Animals were considered fully recovered once they were able to eat and drink, able to ambulate to food and water, and maintain stable vitals (oxygen saturation levels, rectal temperature, heart rate, and respirations per minute). For the remainder of the study until euthanasia, wellness checks were performed twice daily by lab members to observe physical and cognitive status.

### 2.6. Gait Assessment

Injury animals underwent gait assessment once at least 1 day prior to injury, then at 1, 4 and 7 days post-injury, and healthy animals underwent gait assessment on three non-consecutive days. For all animal study days, including both experimental and healthy groups, animal weights were recorded in kilograms and used to calibrate the Tekscan Strideway™ to a similar pressure. After calibration, the animal was placed at one end of the Tekscan Strideway™ mat and encouraged to walk across using techniques described previously. An ELP camera (ELP-USBFHD05MT) with infrared (IR) and light emitting diode (LED) was used to record video of gait assessment trials at an acquisition speed of 250 frames per second (ELP, Shenzhen Ailipu Technology Co., Ltd, Shenzhen, China). A trial was defined as one attempt by the animal to cross the mat. A minimum of three trials was collected per animal per study day. Trials were considered to be unacceptable if any of the following occurred: galloping, pause in ambulation, not ambulating directly towards the opposite end of mat, slipping/sliding, or exceeding 25 s to cross mat.

### 2.7. Data Processing

There were two components to a gait assessment trial: (1) a pressure recording collected by the Tekscan Strideway™ gait mat, and (2) a video recording from a camera mounted at the end of the mat. Both recordings were collected through the Strideway™ software. A trial was considered to be successful and acceptable if the animal crossed the full length of the mat in less than 25 s and did not exhibit any unacceptable ambulatory behaviors described previously (galloping, pausing, slipping, sliding, not ambulating directly to the opposite end of the mat). Gait trials were selected for processing through the review of pressure recordings, video recordings, and observation notes taken during data collection. Trials were also validated by confirming that the video and pressure recordings were synchronized. For each animal on each study day, the first three acceptable and validated trials were selected and imported into the Strideway™ software for data extraction. Parameters extracted by the software included the number of stances, gait time, gait distance, gait velocity, cycles per minute (CPM), cycle time, stance time, swing time, stride time, and stride velocity (Table 2). Gait distance was not examined due to the requirement of all animals to cross the entirety of the mat. A data table was created by the Strideway software containing the parameter averages for each trial collected as well as the averages of the trials combined. Data tables were then exported to Microsoft Excel (Microsoft Corporation, Redmon, WA, USA), where standard deviation and standard error were calculated for individual animal daily averages. The animal daily averages were then combined per experimental group (single, multiple, and sham) with subsequent standard deviation and standard error calculated.

## 3. Statistics and Analysis

All statistical analyses were performed using the SPSS Statistics (IBM, Armonk, NY, USA) software. Of the forty-six animals received, eleven were not able to have data from all study days collected. These animals were not included in the analysis of variance tests (ANOVA) with repeated measures but were used in the reference range validations (naive group) and non-parametric tests (experimental group: single, repeated, sham). Figures were generated using Microsoft Excel.

### 3.1. Healthy Reference Range

First, each parameter was categorized as either a single-value parameter or an individual hoof parameter (Table 2). The single-value parameters were gait time, gait velocity, number of stances, cycle time, and cycles per minute and were represented with one value per trial. The individual hoof parameters were stance time, swing time, stride time, stride velocity, and stride length and were collected for each hoof (left front, right front, left hind, right hind) for each trial.

For the single-value parameters, a one-way ANOVA with repeated measures and a post-hoc Bonferroni were performed in the healthy group to determine if there was an effect of day. Parameters were excluded if there was found to be an effect of day in the healthy group due to the potential for significance found post-injury in the experimental groups to be unrelated to anesthesia or injury. No effect of day was found; therefore, gait time, gait velocity, number of stances, cycle time, and cycles per minute were studied in the experimental groups.

For the individual hoof parameters, a two-way ANOVA with repeated measures and a post-hoc Bonferroni were used in the healthy group to understand if there was an effect of day or hoof. Stance time and stride length were found to have no effect of day or hoof in the healthy group and were studied in the experimental groups. If the day was a significant factor, the parameter was excluded.

In the experimental groups, a three-way ANOVA with repeated measures and a post-hoc Bonferroni were performed on stance time and stride length to understand if hoof had an effect post-injury. Stance time was found to have no effect of hoof post-injury; therefore, the hoofs were averaged for each study day per each animal in both the healthy and experimental groups. Stride length was found to have an effect of hoof post-injury, so hoofs were studied and reported separately for all groups.

In total, gait time, gait velocity, number of stances, cycle time, and cycles per minute (single-value parameters), along with stance time and stride length (individual hoof parameters), were selected for further statistical analyses. The 2.5th and 97.5th percentiles were calculated for each parameter to establish reference range (RR) interval values in piglets that followed a method congruent with reference intervals established for healthy patients in the clinical setting [55]. To establish the reproducibility of the calculated reference ranges, a separate validation group of healthy animals that did not complete all three study days were compared against the ranges (*n* = 6). Animals were studied on one or two days, and data from each day was considered to be a single data point (*n* = 10). The percentage of data points that fell in the healthy range was evaluated to determine whether the reference range was an accurate representation of healthy values. If the percentage of validation data points in the healthy range was below 75%, the parameter was deemed not reliable. All seven parameters satisfied the criteria for reliability, with at least 75% of the validation group falling within the healthy range. These reliable parameters were then evaluated in the experimental groups to determine the percentage of animals that fell within the healthy reference range on each day of study.

### 3.2. Experimental Groups

The influence of experimental group (sham, single, multiple) and study day (pre-injury, Day One, Day Four, Day Seven) were examined for six parameters (number of stances, gait time, gait velocity, cycle time, cycles per minute, and stance time) using a two-way ANOVA with repeated measures, followed by a post-hoc Bonferroni. For stride length, a three-way ANOVA (group, day, hoof) was performed along with a post-hoc Bonferroni. For all evaluations, significance was defined as *p* < 0.05.

The percentage of animals in each experimental group whose trial values fell within the healthy reference range was determined on each study day to determine if there were significant variations between these proportions by experimental group and by day. For all evaluations, significance was defined as *p* < 0.05. To examine if the experimental group had a significant effect, a Fisher Exact Test was performed. If a group was found to be significant, the Fisher Exact Test was then repeated and restricted to comparing two groups against each other for the post-hoc analysis. To determine if study day had a significant effect within an experimental group, a Cochran Q test was performed, along with a post-hoc McNemar’s test.

## 4. Results

### 4.1. Overview

For the pre-injury study day, there was found to be no significant effect of study group (*p* > 0.05) for any of the parameters (Figure 1, Figure 2, Figure 3, Figure 4, Figure 5, Figure 6 and Figure 7, Appendix A). On Day One post-injury, the multiple injury group was found to have significantly increased gait time, cycle time, and stance time, and decreased gait velocity and cadence, relative to the sham group. For all the above parameters except gait time, differences between the multiple injury and sham groups persisted on to Days Four and Seven post-injury. The only significance found for the single injury group was decreased cadence on Day Seven post-injury, and decreased stride length on both hind limbs (left hind and right hind) on Day Four post-injury, relative to the sham group.

Within the multiple injury group, gait time, cycle time, and stance time were found to be significantly increased on Day One post-injury relative to all other study days, and gait velocity was found to be decreased on Day One relative to all other study days. Cadence was only found to be decreased on Day One relative both to pre-injury and Day Four. Number of stances was found to be increased on Day One relative to both Day Four and Day Seven.

Within the single injury group, there was a delayed increase in gait velocity from Day One to Day Seven, and for number of stances, there was a decrease in stances from pre-injury to Day Four and Day Seven, as well as from Day One to Day Seven. Within the sham group, there was an increase in gait velocity from pre-injury to Day Four, and a decrease in stances from pre-injury to Day Four and Day Seven.

When applying the reference range to the groups, a significant decrease in the proportion of animals in the healthy reference range was found in the multiple injury group on Day One relative to Day Four for gait time, gait velocity, cycle time, and cadence. Gait velocity also experienced this decrease on Day One relative to Day Seven. For number of stances, the multiple injury group experienced an increase in the proportion of animals in the reference range from pre-injury to Day Four. For stride length, only the hind limbs in the single injury group experienced a significant proportion of animals outside the reference range on Day Four relative to the sham group.

### 4.2. Gait Parameters

#### 4.2.1. Number of Stances

The number of stances was found to demonstrate no significant differences between groups for all study days. Within the sham group, the number of stances was found to be significantly decreased on Day Four (*p* < 0.001) and Day Seven (*p* < 0.001) compared to pre-injury. Within the single injury group, Day Four (*p* < 0.001) and Day Seven (*p* < 0.001) were found to have decreased stances compared to pre-injury. Day Seven was also found to have decreased stances relative to Day One (*p* = 0.032). Within the multiple injury group, Day Four (*p* = 0.014) and Day Seven (*p* = 0.007) were found to have decreased stances relative to pre-injury values. Day One was found to have increased stances relative to Day Four (*p* = 0.011) and Day Seven (*p* = 0.005). For reference range comparisons, there was no effect of group on any study day; however, it was found that the multiple injury group had an increase in animals in the reference range on Day Four relative to pre-injury (*p* = 0.031) (Figure 1A,B).

#### 4.2.2. Gait Time

Gait time was found to be increased in the multiple injury group compared to the sham group on Day One post-injury (*p* = 0.042). Within the single injury group, Day Four post-injury was found to have decreased gait time relative to pre-injury (*p* = 0.035). Within the multiple injury group, gait time on Day One post-injury was found to be significantly increased compared to pre-injury (*p* = 0.02), Day Four (*p* = 0.015), and Day Seven (*p* = 0.012). For reference range comparisons, the multiple injury group’s proportion of animals in the healthy reference range was significantly decreased on Day One relative to Day Four (*p* = 0.031), with all out-of-range animals exhibiting longer gait times than the healthy RR. (Figure 2A,B).

#### 4.2.3. Gait Velocity

The multiple injury group had decreased gait velocity on Day One (*p* = 0.009), Day Four (*p* = 0.032), and Day Seven (*p* = 0.041) post-injury relative to the sham group. Within the sham group, gait velocity on Day Four had significantly increased (*p* = 0.014) from pre-injury values. Within the single injury group, gait velocity on Day Seven was significantly increased from Day One (*p* = 0.028). Within the multiple injury group, gait velocity on Day One was significantly decreased compared to pre-injury (*p* = 0.031), Day Four (*p* = 0.008), and Day Seven (*p* = 0.023). For reference range comparisons, the multiple injury group’s proportion of animals in the healthy range was significantly decreased on Day One relative to Day Four (*p* = 0.031) and Day Seven (*p* = 0.031), with all out-of-range gait velocities slower than the healthy RR (Figure 3A,B).

#### 4.2.4. Cycle Time

The multiple injury group was found to have significantly decreased gait cycle time on Day One (*p* = 0.030), Day Four (*p* = 0.005) and Day Seven (*p* = 0.005) post-injury relative to the sham group. Within the multiple injury group, cycle time on Day One was significantly decreased compared to pre-injury (*p* = 0.013), Day Four (*p* = 0.017), and Day Seven (*p* = 0.014). The multiple injury group had a significantly decreased proportion of animals in the healthy reference range on Day One relative to Day Four (*p* = 0.031), with all out-of-range cycle times longer than the healthy RR (Figure 4A,B).

#### 4.2.5. Cycles per Minute (Cadence)

The multiple injury group had a significantly decreased cadence on Day One (*p* = 0.003), Day Four (*p* = 0.015), and Day Seven (*p* = 0.006) post-injury relative to the sham group. The single injury group was found to have decreased cadence on Day Seven (*p* = 0.045) post-injury relative to the sham group on Day Seven. Within the multiple injury group, cadence on Day One was found to be decreased relative to pre-injury (*p* = 0.006) and Day Four (*p* = 0.005) post-injury. The multiple injury group had a significantly decreased proportion of animals in the healthy reference range on Day One relative to Day Four (*p* = 0.031), with all out-of-range cadences slower than the healthy RR. (Figure 5A,B).

#### 4.2.6. Stance Time

The multiple injury group stance time values were found to be significantly increased on Day One (*p* = 0.035), Day Four (*p* = 0.046) and Day Seven (*p* = 0.012) post-injury relative to the sham group. Within the multiple injury group, stance time was significantly increased on Day One relative to pre-injury (*p* = 0.035), Day Four (*p* = 0.021) and Day Seven (*p* = 0.022) post-injury. No significant differences were found for reference range comparisons (Figure 6A,B).

#### 4.2.7. Stride Length

In both hind limbs (left hind; LH and right hind; RH), the single injury group was found to have significantly decreased stride length on Day Four post-injury (*p* < 0.017) relative to the sham group. However, no significant differences were found between groups in either front limb (left front; LF and right front; RF, Figure 7A–D).

In all four limbs, the sham and single injury group experienced an increase in stride length from pre-injury to Day Four (*p* ≤ 0.001) and Day Seven (*p* ≤ 0.001). Within the single injury group, we also noted a significant increase in stride length of all limbs on Day One (*p* ≤ 0.016) and Day Four (*p* < 0.04) relative to Day Seven (Figure 7A–D).

In all four limbs, the multiple injury group experienced an increase in stride length from pre-injury to Day Four (*p* ≤ 0.007). Additionally, in the multiple group a significant increase from pre-injury to Day Seven (*p* ≤ 0.031) was observed in all limbs except for the right front (RF). Only the right hind (RH) limb experienced a significant decrease in stride length from pre-injury to Day One (*p* = 0.029) in the multiple injury group. All four limbs also displayed an increase in stride length on Day Four (*p* ≤ 0.003) and Day Seven (*p* ≤ 0.002) relative to Day One (Figure 7A–D).

No significant differences were found for reference range comparisons in either front limb, i.e., left front (LF) and right front (RF). However, in both hind limbs, i.e., left hind (LH) and right hind (RH), the single injury group had a decreased proportion of animals in the reference range relative to the sham group (*p* = 0.018) (Figure 7E–H).

## 5. Discussion

### 5.1. General Summary

In this study, we identified persistent changes in gait patterns following a sagittal RNR injury and the exacerbating influence of multiple head rotations at Day Onepost-injury. Our findings suggest that at Day One post-injury, our multiple RNR group walked significantly slower (higher gait time and lower velocity), had fewer step cycles per minute (lower cadence), and spent longer time with their feet on the ground (higher cycle time and stance time). Additionally, in our multiple RNR group, velocity, cycle time, cadence, and stance time were not only affected at Day One post-TBI, but deficits in these parameters significantly persisted at Day Four and Day Seven post-injury, suggesting that multiple injuries have long-term effects on gait. Similarly, the literature shows that children walk slower, take fewer steps per minute, and have difficulty maintaining balance months or years post-injury [7,9,56,57,58]. In contrast, multiple RNR injury did not seem to significantly increase the total stances (number of stances) or shorten the distance between steps (stride length). While we did not notice a significant persistent reduction in the distance between steps (stride length), studies in children have identified significant reductions in stride length and shorter step length caused by TBI [7,9,10]. These findings in children highlight the long-term effect that TBI has on gait, which we also found in our piglet multiple RNR group but with different gait parameter alterations following TBI, as discussed above.

### 5.2. Relationship with Previous Pediatric Studies

Children who sustain moderate to severe TBI show evidence of decreased velocity that persists for years post-injury (Appendix A). Two studies carried out 3–12 months after brain injury in young children indicated a 27.7% and 20% decrease in velocity [9,59]. Another study noted a 50% reduction in gait speed 3.5 years after severe TBI in adolescents [10]. Based on our findings, our multiple RNR group also showed a 50% significant reduction in velocity at Day One post-injury and maintained a 24% significant decrease in velocity by Day Seven post-injury compared to sham animals. We noted that at Day Seven post-injury, the percentage decrease in velocity observed in our multiple RNR group was similar to those documented in young children by 1 year post-injury. This significant reduction in the velocity of our multiple RNR group may be due to the piglets attempts in maintaining balance and stability, as noted in pediatric TBI and older adult studies [14,59,60].

Studies in pediatric TBI have also identified decreases in cadence that persist for up to 7.8 months following moderate to severe TBI (Appendix A). At 2.8 months post-injury, a TBI study in young children noted a 13% decrease in cadence, and although there was an improvement by 7.8 months post-injury, a 5.6% decrease in cadence persisted [7]. This study also noted a 23% and 7.1% decrease in velocity at 2.8 months and 7.8 months post-injury, respectively. The percentage decrease observed in young children at both time points were quite similar for velocity and cadence [7]. Compared to the sham group, our single RNR group exhibited a 14% significant decrease in cadence at Day Sevenpost-injury. Our multiple RNR group also displayed a 45% significant decrease in cadence at Day One post-injury, and a 20% significant decrease in cadence by Day Seven post-injury compared to sham animals. Similar to observations in young children, the decrease in cadence was also quite similar to the decrease in velocity in our multiple RNR group. It is important to note that velocity is a product of cadence and stride length, and it can be significantly affected by changes in either one of these parameters [61]. The similarity observed in changes to cadence and velocity enables us to conclude that multiple RNR injury may not have an effect on stride length, and this is reflected by the non-significant changes observed in stride length for this group. However, studies in children are quite different from our findings because they show that velocity, cadence, and stride length seem to decrease and improve together post-injury [7]. These differences may be due to children being bipedal, and piglets being quadrupedal, which is a distinction that should be put into consideration when assessing gait patterns from both species. Cadence has also been shown to alter balance, and the significant decrease in cadence in our multiple RNR group may be responsible for the difficulty that this group experienced in maintaining balance during gait trials [62].

Variability in gait patterns and decrease in stride length are other impairments that have been identified in children and piglets post-TBI [37,63,64,65]. A pediatric TBI study noted increased step variability in children who suffered severe TBI [10]. Another study showed that despite significant improvements by 7.8 months post-injury, differences in stride length were still present in children who had suffered moderate to severe TBI compared to those who had not [7]. This study noted a 16% decrease in stride length at 2.8 months post-injury, with a slight improvement to about 7.9% decrease by 7.8 months post-injury. Kinder and colleagues also noted a decrease in hind reach in their pediatric controlled cortical impact (CCI) piglet model and described it as the lagging behind of the hind limbs compared to the front limbs [36]. They proposed that this decrease in hind reach may be caused by an overall decrease in stride length and an increase in percentage stance. Our single RNR injury group had significantly shorter stride length in the left and right hind limbs at Day Four post-injury compared to the sham group. This significant difference was not observed in the front limbs, which may signify some level of variability in gait patterns of the front (LF, RF) and hind (LH, RH) limbs of this group. Although noted at Day Four post-injury in the single RNR group, we did not identify gait variability in the front and hind limbs of the single RNR group at Day One and Seven post-injury and in the multiple RNR group at Days One, Four, and Seven post-injury compared to the sham group. Recent studies in children and piglets also mention that a decrease in hind reach and stride length may be responsible for directly influencing an increase in cycle time, decrease in cadence, and decrease in velocity [36,66,67]. Our multiple RNR group displayed a 152% significant increase in cycle time at Day One post-injury, and a 30% significant increase in cycle time by Day Seven post-injury. Although we noted a significant increase in cycle time and significant decreases in cadence and velocity, no decrease in stride length was observed in this group, perhaps due to a concurrent significant group-independent weight gain from pre-injury (7.11 ± 0.6 kg mean ± SE) to Day Seven (9.33 ± 0.6 kg, 2-way ANOVA with repeated measures) (Table 3).

Additionally, although several studies indicate that children with TBI improve post-injury, many of these studies also show significant differences in velocity, cadence, and stride length months or years post-injury compared to healthy controls [7,9,10,59,68]. Kuhtz-Buschbeck and colleagues noted that velocity and cadence improved in about 80% of pediatric patients, but differences between controls and injured children persisted months post-injury [7]. An improvement of 67% in gait time, 83% in velocity, 54% in cycle time, 54% in cadence, and 55% in stance time were observed in our multiple RNR injury group by Day Seven post-injury compared to Day One post-injury. However, irrespective of these improvements in gait, significant differences in gait time, velocity, cycle time, cadence, and stance time persisted by Day Seven post-injury. Similarly, a focal TBI piglet study also noted transient impairments in their cycle time, cadence, and stride length [36]. Significant differences in cycle time and cadence of their CCI piglets persisted by Day Seven post-injury compared to their baseline measures. These findings suggest that our sagittal RNR piglet models exhibit similar gait abnormalities as seen in both humans and CCI piglet models during the early and later phases post-injury.

In summary, this is the first study to (i) utilize a piglet diffuse TBI model to evaluate changes in gait patterns, (ii) compare gait changes in injured groups to both healthy and sham groups, and (iii) evaluate changes in gait patterns following repeated mild TBI.

### 5.3. Limitations and Future Work

A distinction of this diffuse TBI study is that the levels of rotational head loads applied to the piglets are associated with active sport participation in heading the ball in soccer and with common recreational behavior in children. In previous studies published by our group as a model of abusive head trauma in infants, younger 3–5-day old piglets (35–40 gm brain, typically) experienced single or repeated cyclical “trains” of sagittal head rotations with velocities of 20–40 rad/s and accelerations of 600–700 rad/s^2^ scaled from reconstructions of human infant shaking [69,70]. Using traditional mass-scaling laws [71] to determine equivalent kinematics in a 4-week old piglet used in the current study (60 gm brain, typically), the equivalent rotational loads for vigorous shaking correspond to 15–35 rad/s and 450–530 rad/s^2^. Based on these scaled levels, the mTBI loads in the current study for the “median headers” were three times higher rotational velocities and 30 times higher rotational accelerations than in vigorous shaking (Table 1). While we observed prolonged gait deficits following head rotations at levels associated with recreational play in children, future work should expand on the functional deficits associated with much lower level head rotations indicative of shaking without impact. Another limitation of this study was the use of only female piglets in our experimental group. Gait analysis in both sexes is needed to better understand sex variability in gait impairments. In humans, women tend to report longer and worse outcomes than men, yet pre-clinical studies [72] show females subjects are known to have lower comorbidities, implicating female sex hormones may have neuroprotective effects [31]. This limitation was mitigated by using pre-pubertal female piglets. A third limitation is the use of only a sagittal RNR injury model. Additional studies in axial and coronal directions should be explored to provide a more complete understanding of how different injury directions can affect motor function. Lastly, another limitation was the exclusion of cognitive impairments and neuropathological assessments which may affect gait. In future studies we intend to study both sexes, examine various injury directions, study cognitive impairment, quantify neuropathology, and observe changes in gait impairments for longer periods of time to ensure that there are no further declines after Day Seven post-injury.

## 6. Conclusions

In conclusion, we observed that sagittal RNR injury can lead to significant acute increase in gait time, decrease in velocity, decrease in cadence, shorter stride length, increase in stance time, and increase in cycle time, much like pediatric TBI patients. Multiple RNR injury was observed to cause worse gait impairments compared to single RNR injury, and multiple RNR injury metrics were significantly outside the healthy reference range. Based on the similarities between our findings and pediatric TBI studies, as well as the anatomy, development, and size of a piglet and a child’s brain, these results indicate that a sagittal RNR piglet model can serve as an objective quantitative functional platform in the understanding and treatment of gait impairments due to TBI using novel therapeutics.

## Figures and Tables

**Figure 1 biomedicines-10-02976-f001:**
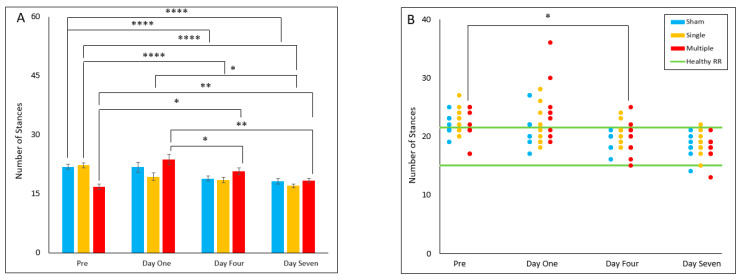
Number of Stances (**A**) Presents daily group average ± standard error and statistical comparisons noted from the two-way ANOVA with repeated measures analysis; (**B**) Presents individual animal daily averages and statistical comparisons are noted from the Fisher Exact Test and McNemar’s analysis (* *p* < 0.05, ** *p* < 0.01, **** *p* < 0.001).

**Figure 2 biomedicines-10-02976-f002:**
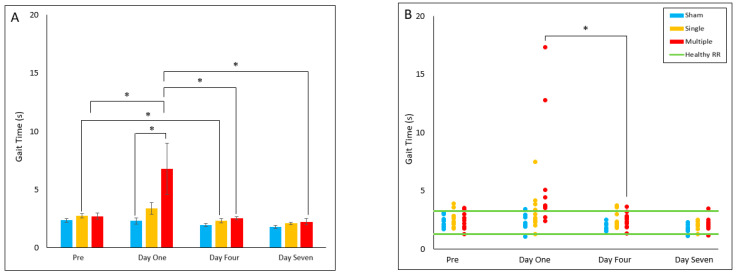
Gait Time (**A**) Presents daily group average ± standard error and statistical comparisons noted from the two-way ANOVA with repeated measures analysis; (**B**) Presents individual animal daily averages with statistical comparisons noted from the Fisher Exact Test and McNemar’s analysis (* *p* < 0.05).

**Figure 3 biomedicines-10-02976-f003:**
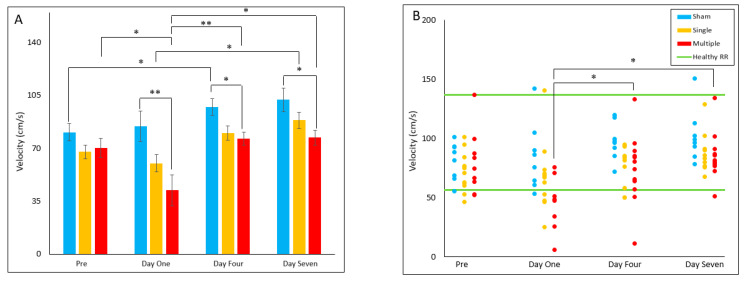
Gait Velocity (**A**) Presents daily group average ± standard error and statistical comparisons noted from the two-way ANOVA with repeated measures analysis; (**B**) Presents individual animal daily averages with statistical comparisons noted from the Fisher Exact Test and McNemar’s analysis (* *p* < 0.05, ** *p* < 0.01).

**Figure 4 biomedicines-10-02976-f004:**
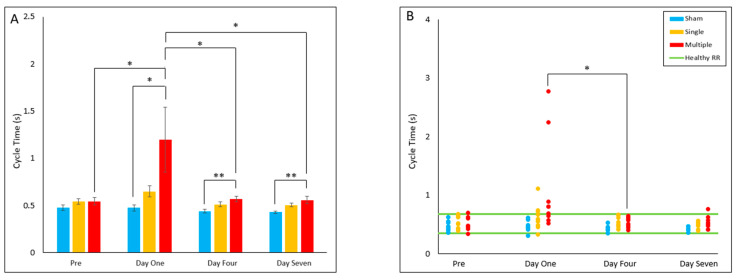
Cycle Time (**A**) Presents daily group average ± standard error and statistical comparisons noted from the two-way ANOVA with repeated measures analysis; (**B**) Presents individual animal daily averages with statistical comparisons noted from the Fisher Exact Test and McNemar’s analysis (* *p* < 0.05, ** *p* < 0.01).

**Figure 5 biomedicines-10-02976-f005:**
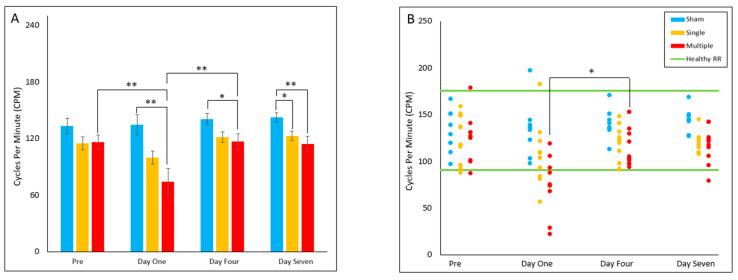
Cycles per Minute (cadence) (**A**) Presents daily group average ± standard error and statistical comparisons noted from the two-way ANOVA with repeated measures analysis; (**B**) Presents individual nimal daily averages with statistical comparisons noted from the Fisher Exact Test and McNemar’s analysis (* *p* < 0.05, ** *p* < 0.01).

**Figure 6 biomedicines-10-02976-f006:**
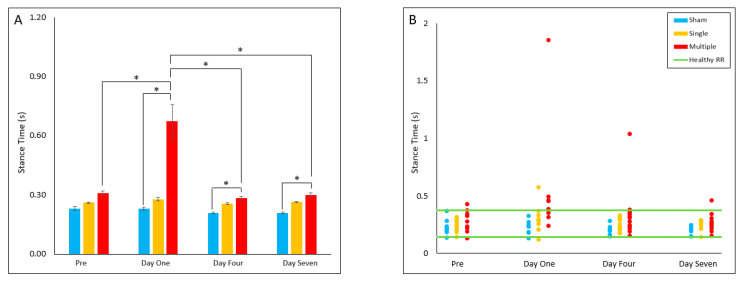
Stance Time (**A**) Presents daily group average ± standard error and statistical comparisons noted from the two-way ANOVA with repeated measures analysis; (**B**) Presents individual animal daily averages with statistical comoparisons noted from the Fisher Exact Test and McNemar’s analysis (* *p* < 0.05).

**Figure 7 biomedicines-10-02976-f007:**
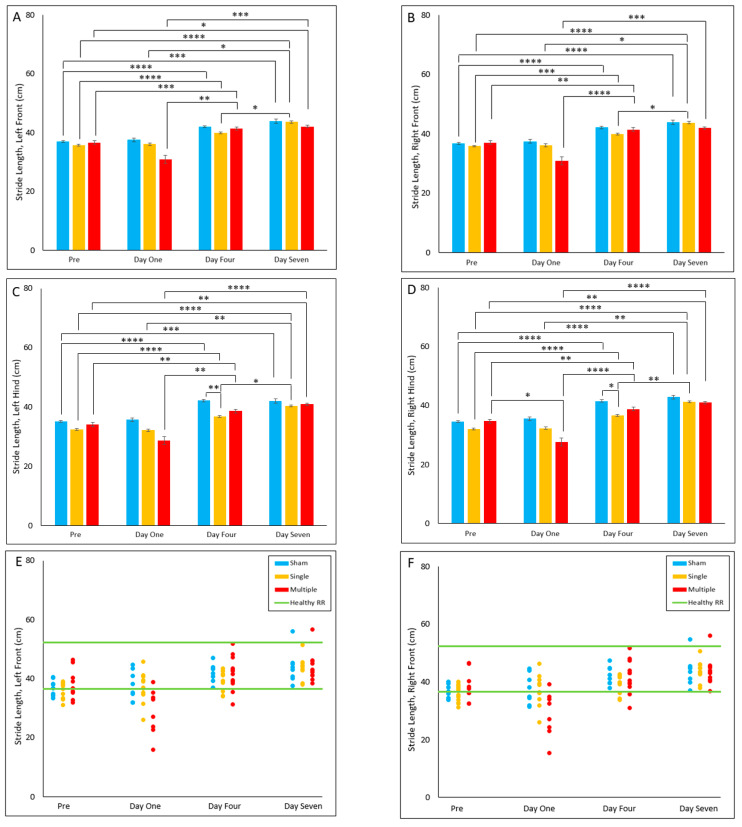
Stride Length by Limb (**A**–**D**) Presents daily group average ± standard error and statistical comparisons noted from the two-way ANOVA with repeated measures analysis; (**E**–**H**) Presents individual animal daily averages with statistical comparisons noted from the Fisher Exact Test and McNemar’s analysis (* *p* < 0.05, ** *p* < 0.01, *** *p* = 0.001, **** *p* < 0.001).

**Table 1 biomedicines-10-02976-t001:** Summary of Angular velocity and accelerations of single and multiple RNR injuries. Values represent the calculated average ± standard error.

	Rotation Type	Angular Velocity (rad/s)	Angular Acceleration (rad/s²)
Single	High	104.5 ± 0.47	40,052 ± 1559
Multiple	Median	61.3 ± 0.18	15,010 ± 169
	High	104.6 ± 0.41	38,368 ± 499

**Table 2 biomedicines-10-02976-t002:** Parameter definitions.

Parameter Type	Parameter	Definition
Single-Value Parameters	**Number of Stances**	Also known as ‘Number of Strikes’; how many total stances an animal makes during trial; stances in quadrupeds can involve 2- or 3-legged support
	**Gait Time**	Time, in seconds, that it takes for animal to cross the gait mat; begins with contact of the first left or right front stance and ends with the time of contact of the last left or right front stance registered on the sensor
	**Gait Velocity**	Gait distance divided by gait time; centimeters per second
	**Cycle Time**	Average time, in seconds, to complete a gait cycle
	**Cycles Per Minute**	Also known as “cadence”; number of complete gait cycles per minute
Individual Hoof Parameters (left front, right front, left hind, right hind)	**Swing Time**	Elapsed time between the last contact of a preceding hoof and first contact of the next step of that same hoof, in seconds
	**Stride Time**	Elapsed time between the first contacts of two consecutive hoof falls, in seconds
	**Stance Time**	Average time from first contact to last contact of each hoof, in seconds
	**Stride Length**	Distance measured parallel to the line of progression, between the posterior heel points of two consecutive hoof falls, in centimeters
	**Stride Velocity**	Stride length divided by stride time for each hoof; centimeters per second

**Table 3 biomedicines-10-02976-t003:** Summary of animal weight averages (in kilograms) for pre-injury and Day Seven post-injury and analysis results. Values represent the calculated group average ± standard error and the *p*-value for Paired-Sample *t*-test.

	*Pre-Injury*	*Day Seven*	*p-Value*
Sham	6.34 ± 0.601	9.20 ± 0.728	<0.001
Single	6.60 ± 0.221	8.90 ± 0.444	<0.001
Multiple	7.11 ± 0.556	9.33 ± 0.625	<0.001

## Data Availability

Not applicable.

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
