# Peer review of "Multiple Head Rotations Result in Persistent Gait Alterations in Piglets"

_biomedicines, 2022, doi:10.3390/biomedicines10112976_

Round 1
Reviewer 1 Report
This paper refers to a pediatric head injury model. This study is very important because pediatric head trauma has many issues that need to be improved.
a) What is the reason for selecting pigs as model animals? Large animals are not recommended in recent animal research regulations.
b) Please provide the specific mechanism that caused this gait disturbance. For example, present the damaged area using MRI, immunostaining of brain parenchyma, etc.
c) The authors measured up to 7 days after trauma, can we evaluate this as equivalent to the chronic phase of head injury in human terms?
d)The memory and title text in Figure is too small and very difficult to see. Please improve.
Author Response
Reviewer 1 comments/suggestions: This paper refers to a pediatric head injury model. This study is very important because pediatric head trauma has many issues that need to be improved.
Thank you for your thoughtful comments; they have enriched the manuscript greatly. We have addressed each comment and question below, and indicate where we have incorporated suggested changes in the manuscript.
- a) What is the reason for selecting pigs as model animals? Large animals are not recommended in recent animal research regulations.
The porcine model is used because (1) swine have a gyrencephalic brain, like humans, and (2) they have a similar brain development when compared to humans. With both of these facts combined, swine present an optimal model for studying pediatric traumatic brain injury (TBI). Additionally, there have been multiple published studies that support the use of gait in a large animal model, such as swine, to study neurological disease. These explanations and references can be found throughout the “Introduction” section of the manuscript (Line 54-86).
- b) Please provide the specific mechanism that caused this gait disturbance. For example, present the damaged area using MRI, immunostaining of brain parenchyma, etc.
The scope of the research study was to examine head loading conditions consistent with head impact exposures in soccer games in high school athletes. Using our inter-species scaling techniques, we identified the head kinematics that correspond to head impacts in soccer for use in this large animal study. We did not have any clinically diagnosed concussion in the high school soccer players from which we used their head kinematics and thus we do not expect substantial damage to the brain tissue due to the mild severity nature of the head impacts used for the design of this animal study. While providing a specific mechanism to quantify the tissue pathology would be interesting and potentially add another layer of analysis to our work, this specific study did not perform any histopathology or imaging. For future work, we agree that imaging could be helpful in identifying and tracking tissue trauma post-injury at these levels of head kinematics as well as using histopathological analysis post-mortem, and we have added it in the study limitations.
- c) The authors measured up to 7 days after trauma, can we evaluate this as equivalent to the chronic phase of head injury in human terms?
Literature examining the relationship between human and swine brain development suggests many similarities such as a growth spurt occurring around time of birth (Dickerson and Dobbing, 1967), the prepubescent developmental changes in EEG, brain composition and cerebral blood flow (Duhaime, 2006), and the white matter maturation throughout adolescence (Ryan, 2019). Comparing these developmental rates to maturity in the piglet at 4.5-6 months to those in humans, the 7-day survival in our piglets would represent a longer period, likely months to a year, in a human. Due to the imprecision of this estimate, we did not include it in our manuscript.
We have elected to include the following references in the manuscript on Lines 71 and 73 to provide further evidence of the similar brain development trajectories between swine and humans. Duhaime (2006) is now reference [35]. Dickerson and Dobbing (1967) is now reference [39], and Ryan (2018) is now reference [40]. All subsequent reference numbers have been revised to match.
d)The memory and title text in Figure is too small and very difficult to see. Please improve.
Figures 1-7, the figure legend and x- and y-axis titles now have increased font size.
Reviewer 2 Report
This study has examined the impact of single and multiple head rotations on gait in piglets, which aims to model traumatic brain injury in children. The study has provided value functional outcome measures following single and multiple rapid non-impact head rotations in the piglet, which will be useful for the future evaluation of potential therapeutics and interventions to improve outcomes after single and repeated TBI.
The manuscript is well written manuscript, and study is well executed. The authors also do an excellent job discussing how their results relate to the findings of previous studies.
In Abstract is it worth mentioning the number of RNRs for multiple injury model?
Line 182-183: “..experimental group animals were studied for one day pre-injury and three days post-injury.” - Suggest rewording last part of sentence to “..experimental group animals were studied for one day pre-injury and on days 1-, 4- and 7 post-injury.”
Line 315: “(p > .05)” change to “(p > 0.05)”. Correct throughout manuscript.
Line 342: “The number of stances was found to have no effect of group across all study days”- not sure what this sentence means??
359: Within the single injury group, Day +4 post-injury was found to have increased gait time relative to pre-injury (p = .035). It doesn’t look like it to me in Figure 2A.- should it be Day +1 post-injury??
Discussion: Summary section contains more than a summary of study. Consider dividing into three paragraphs/parts; 1. Discussion of rotational head loads, 2. Discussion on limitations of study, and 3. Summary.
Summary could be combined with Conclusions section.
Author Response
Reviewer 2 comments/suggestions: This study has examined the impact of single and multiple head rotations on gait in piglets, which aims to model traumatic brain injury in children. The study has provided value functional outcome measures following single and multiple rapid non-impact head rotations in the piglet, which will be useful for the future evaluation of potential therapeutics and interventions to improve outcomes after single and repeated TBI. The manuscript is well written manuscript, and study is well executed. The authors also do an excellent job discussing how their results relate to the findings of previous studies.
Thank you for your thoughtful comments; they have enriched the manuscript greatly. We have addressed each comment and question below, and indicate where we have incorporated suggested changes in the manuscript.
In Abstract is it worth mentioning the number of RNRs for multiple injury model?
We have added a short note to Line 14, “following a single and multiple (x5) sagittal rapid non-impact head rotation (RNR) to convey that the multiple rotation group experienced exactly 5 rotations.
Line 182-183: “..experimental group animals were studied for one day pre-injury and three days post-injury.” - Suggest rewording last part of sentence to “..experimental group animals were studied for one day pre-injury and on days 1-, 4- and 7 post-injury.”
Lines 184-185 now reads as “Healthy animals were studied for three non-consecutive days, and experimental group animals were studied for one day pre-injury and on 1-, 4-, and 7-days post-injury.”
Line 315: “(p > .05)” change to “(p > 0.05)”. Correct throughout manuscript.
We have now changed all p-value references to have a “0” prior to the p-value decimal.
Line 342: “The number of stances was found to have no effect of group across all study days”- not sure what this sentence means??
The sentence should explain that there was no significant difference found between the sham, single, and multiple groups on Pre-Injury, 1-, 4- and 7-days Post-injury. This sentence has now been revised on Line 346-347, “The number of stances was found to have no significant differences between groups for all study days”.
359: Within the single injury group, Day +4 post-injury was found to have increased gait time relative to pre-injury (p = .035). It doesn’t look like it to me in Figure 2A.- should it be Day +1 post-injury??
After further review of our gait time analysis results, Day +4 was significantly decreased relative to Pre-injury within the single injury group. The correction has been made on Line 367, “Within the single injury group, Day +4 post-injury was found to have decreased gait time relative to pre-injury (p = 0.035)”.
Discussion: Summary section contains more than a summary of study. Consider dividing into three paragraphs/parts; 1. Discussion of rotational head loads, 2. Discussion on limitations of study, and 3. Summary.
Dividing the summary improves the overall structure and flow of the revised manuscript. It is now composed of the following three parts: (1) General summary, (2) Relationship with previous pediatric studies, and (3) Limitations and Future Work.
Summary could be combined with Conclusions section.
After consideration, we have elected to keep the Summary and Conclusions section separate to distinguish our discussion of the results in the context of the literature from our concluding messages.
Round 2
Reviewer 1 Report
Dear, authors
Thank you for your revision.
I understand that your study is based on other reports and your theoretical development. However, I do not trust it simply because head trauma caused gait disturbance. You need to provide specific evidence that is clinically proven, for example, damage to specific neural pathways or accumulation of tau protein.